# Use of Infrared Thermography to Assess Body Temperature as a Physiological Stress Indicator in Horses during Ridden and Lunging Sessions

**DOI:** 10.3390/ani12233255

**Published:** 2022-11-23

**Authors:** Joana Noronha Martins, Severiano R. Silva

**Affiliations:** 1Animal Science Department, University of Trás-os-Montes e Alto Douro, Quinta de Prados, 5000-801 Vila Real, Portugal; 2Veterinary and Animal Research Centre (CECAV) and Associate Laboratory of Animal and Veterinary Science (AL4AnimalS), University of Trás-os-Montes e Alto Douro, Quinta de Prados, 5000-801 Vila Real, Portugal

**Keywords:** horse, welfare, workload, personality, infrared thermography

## Abstract

**Simple Summary:**

Equitation is based more on traditions than science. Current training methods might be efficient in the short term but lack scientific evidence that disproves any undesired effects they might have on the horse, both physical and emotional. This study aims to test easy-to-use technology (infrared thermography) for testing the fitness of dressage horses during ridden work, riding lessons and lunging and to find a relationship between the horse’s personality and its reaction to stress during exercise. The behavior during exercise was shown to reflect personality, sex and age, but thermographic readings exhibited no variation. Although the technology was not proven useful with dressage horses training in a familiar environment, its usage in combination with other tools is yet to be tested in such conditions. Despite personality affecting behavior, it did not have a relationship with the level of stress on the horse caused by exercising. Testing this technology can lead to the creation of efficient training programs, safeguarding horses from over- and underworking.

**Abstract:**

Equitation is a cause of physiological stress in the equine athlete, and personality is a factor generally associated with the different responses of equines to stressors. This study explored ocular temperature, measured via infrared thermography, associated with personality and stress in horses submitted to dressage exercising in riding lessons, ridden training and lunging. Infrared thermograms of 16 horses were taken before and after sessions using an FLIR F4 camera (FLIR Systems AB, Sweden) to determine maximum eye surface IRT temperature (IRTmax), and total training time was registered (T). A novel-stimulus test was conducted for personality assessment, and the ridden behavior was scored (mRBS). The results showed that T was statistically different (*p* ˂ 0.001) between modalities, but no differences were found in any IRTmax tests. Statistical correlations were found between mRBS and personality groups, sex and age (all *p* < 0.001). Additionally—and with caution, given the sample—no association was found between mRBS and post-workout IRT readings and modality, or between pre-workout IRT readings and personality groups. We conclude that trained horses show little stress when working in a familiar environment and when the workout plan is submaximal. The personality test was adequate and positively correlated with ridden behavior.

## 1. Introduction

The anthropomorphization of horses has led to a wide variety of methodologies to be used in handling this animal. Traditions, personal beliefs and experiences distinguish equestrians [1] and their techniques; these frequently exhibit apparent success, but do not consider how a horse learns [2]. A technique’s success depends on the animals’ life experiences, temperament and learning ability [2]. Misunderstanding of these concepts and how training impacts welfare leads to wrongfully seeing behavioral issues as being derived from personality rather than poor handling [3,4]. Personality is the result of the effects of life experiences and environment on temperament [5], while temperament is defined as stable behavioral tendencies present in early life [6], although the terms have been used interchangeably. As with humans, personality affects the response to stimuli, namely during handling [7,8,9,10], horse–object/investigative interaction [11,12] and social interaction [8]. A generalized personality type can be attributed based on direct observation, measurement of behavioral and physiological markers or a combination [13]. For example, novel-stimulus tests study reactivity as a trait of equine personality, frequently scoring defined behavioral markers on a scale [13]. 

Poor fitness and how to assess it is an issue across all equines, from the ill-prepared athlete [14,15,16] to the often overweight leisure ride [17,18,19] or the apathetic working equine [19]. An unfit horse is at higher risk of injury [14,15,16], leading to early interruption in training, wastage and costlier veterinary fees [14]. Heart rate (HR), ventilation rate (VR), and blood and plasma lactate concentration are the most common biomarkers correlated with assessing fitness in field conditions [20,21,22,23,24]. In fact, HR-measuring technologies are accessible to riders through commercial products such as girth attachments with electrodes. These markers can be incorporated into standardized exercise tests or tests with gradual increments of effort, looking for changes in readings over time [20]. During exercise, chemical energy is converted into mechanical energy with an efficiency rate of 20% [25,26]. Therefore, 80% of the energy is lost as heat, enough to raise body temperature by 3 to 5 °C [27]. To regulate core body temperature, the horse possesses mechanisms of dissipation through conduction, convection and radiation [25,27]. Sweat evaporation represents between 50 and 75% of heat dissipation [25,26], while surface radiation and convection account for 9 to 13% in mild weather (<20 °C) or the totality of heat dissipation in cold weather (<10 °C) [26]. Losses through radiation can be captured using infrared thermography (IRT) technology [28] and to determine effort; in theory, as exercise intensity increases, so do metabolic heat production and, therefore, the need for heat loss [25,29]. Additionally, infrared thermography has been tested as a tool for fitness evaluation, whereby correlations were found between: increased body surface temperature and blood lactate concentrations in racehorses [30]; increased eye surface temperature and the enzymatic activity of creatine kinase in ranch horses [31]; and increased eye surface temperature and ventilation rate in racehorses [32]. The head is a ‘hot spot’ [33], and the eye, specifically, is a prime area for IRT reading given its high vasculature [10,31,33], its tendency to register a positive response to exercise [31,32,34,35], and the fact that it is part of an extremity structure, primarily free from fur, experiences little interference from the rider and tack [36], and is easy to image.

In this study, 16 horses were ridden and lunged as part of their daily routines. Possible changes in eye surface temperature were tested via infrared thermography imaging of the eye area before and after every workout session. An association between personality and behavior was established and compared with the thermographic results to attest to any effect it might have on workload. It was hypothesized that: (1) lunging is a form of lower-impact training; (2) long, slow-paced sessions and shorter, fast-paced sessions were comparable, and (3) personality affects ridden behavior, which, in turn, affects workload.

## 2. Materials and Methods

### 2.1. Studied Population

The studied population consisted of 16 horses, mostly Lusitano or crossbred, aged between 5 and 20 years (mean 11.4 ± 4.2 years). With regard to sex, 7 were mares, 5 were stallions and 4 were geldings. All were privately owned leisure horses housed in a riding center, trained in dressage and ridden on a daily or weekly basis. All horses had been at the stables for over a year. The horses were housed indoors in individual 3 × 3 m stalls, with natural ventilation and continuous access to water. Hay was distributed during the day to allow constant availability, and concentrate feeding took place three times daily, dosed individually. The area designated for grooming, tacking and bathing was covered and protected from the elements. All horses had up-to-date vaccines, deworming and shoeing of all four hooves.

### 2.2. Personality Assessment

To attribute personality types, a novel-stimulus test was performed twice on random days. This was carried out in a familiar environment and indoors to avoid arousal from environmental stressors. Standing outside each stall, a camera was held at 1.50 m for 5 min, and then, for another 5 min, the camera (stimulus) was held above the head. There was no interaction or communication with the horse during the test and no movement of the object. Using the focal observation method for one minute, after one minute from the moment the object was raised above the head, a reactivity score was given based on the behavior scale (Table 1). Each score represented a personality type: a score of 1 was indifferent or curious, 2 was reluctant and 3 was reactive. More detail on the creation of the behavior scores can be found in the appendices (Table A1).

### 2.3. Ridden Behavior Analysis

The behavior of the exercising horse was evaluated based on the observation method suggested by Ellis et al. [8], adapted to consider discipline change (from show jumping to dressage) and time of focal observation. Rather than assessing the approach to jumps, the “exercises” were lateral movements, halt and rein back, opening/closing the gate, ground poles, pirouettes and flying change of lead. Additionally, the focal observations were performed at minutes 7, 15 and 25, approximating the start, middle and end of the sessions. A score of 1 to 6 on the ridden behavior scale was given at each observation point, and the mean of the three scores of each session (mRBS) was calculated to use during statistical analysis [8]. The ridden horse behavior scale is depicted in Table 2.

### 2.4. Workload Assessment

The horses were exercised in three modalities: lessons (L), progression training (PT) and lunging (Lu). Regardless of modality, for sessions to be valid, they required walking, trotting and canter on both leads. During lessons, five students of the center rode a selected group of horses under supervision and guidance, consisting of basic gait training and simple exercises (lateral movements, halt and rein back, and ground poles). In progression training, all horses were ridden by the two staff riders and were required to perform exercises with higher frequency and demand. Finally, during lunging sessions, the horses were exercised on a lunge line (about 15 m in diameter) by the same staff rider, using only a lunging cavesson and no lunging aids. Because this study was conducted at a commercial school for a period of two months, little interference was permitted that would change the schedules of customers. However, student riders were not assigned one horse in particular (even those who owned a horse); rather, they would rotate riders across lessons as per the decision of the trainer. Therefore, rider–horse pair randomization was introduced in such manner.

The tacks used in ridden sessions were identical: dressage saddle, bridle with single curb mouthpiece, leg protection, saddle pads and gel half pads. Total training time (T) was registered, as were cool-down times. The term ‘cool down’ describes the actions taken by the rider to allow the horse to gradually recover after sessions prior to dismounting. Thermographic images were taken prior to and after each workout session using an infrared camera (FLIR-F4) emissivity of Ɛ0.95. The camera was placed perpendicularly to the eye at a 1 m distance. Distinction between the left and right eye was not made. The area of imaging encompassed most of the head (Figure 1a), and to determine maximum eye surface temperature (IRTmax), an ellipse was fitted to the eye [10,31] using the ellipse tool of the FLIR Tools+ software (Figure 1b).

Pre-workout images were taken after the horse was restrained with a halter, and the eye area was cleaned prior to tacking. Post-workout images were taken when the horse arrived at the designated imaging area, and the bridle was switched to a halter. Because the sessions and imaging were performed outdoors, the following meteorological data were gathered: minimal and maximal temperature (in Celsius degrees) and wind speed (in kilometers per hour). The mean air temperature was 16.3 ± 3.8 °C, the lowest registered temperature was 10 °C on one occasion, and it also reached a maximum of 24 °C once. Median air temperature was 15.5 °C. The mean wind speed was 18.3 ± 5.3 km/h and the median was 10.5 km/h. The occurrence of days with strong gusts was 29%.

### 2.5. Statistical Analysis

The analyzed data consisted of total training time (T), maximum eye surface temperature (IRTmax), mean IRT maximum eye surface temperature for each horse in each session before (mIRTmax_b) and after a session (mIRTmax_a), and mean ridden behavior score (mRBS). Cool down time, mean air temperature and wind speed were considered for discussion. All numeric data were tested for normality using the Shapiro–Wilk test; none fit a normal distribution. The horses were grouped according to modality when studying workload and personality type to study behavior.

#### 2.5.1. Personality Assessment

The correlation between personality groups and mRBS was tested using the Spearman Rank Correlation test. Variances in mRBS between modalities were assessed using the Kruskal–Wallis H test. The correlation between mRBS and mIRTmax_a was tested using the Spearman Rank Correlation test. The mIRTmax_b variation between personality groups was assessed using the Kruskal–Wallis H test. Finally correlations between sex, age, mRBS and personality were assessed through multiple Spearman Rank Correlation tests, followed by Kruskal–Wallis H tests with post hoc analysis to assess differences between sexes and age groups (5 to 10 years and 11 to 20 years).

#### 2.5.2. Workload Assessment

Total training time between modalities was compared using a Kruskal–Wallis H test to attest for differences in distribution, followed by a post hoc pairwise comparison. The variation in IRTmax before and after a workout of each modality was compared using the Wilcoxon signed-rank test. The pre- and post-exercise IRTmax values between modalities were tested using the Kruskal–Wallis H test for mean ranks.

## 3. Results

### 3.1. Personality Assessment

There was a correlation between personality groups and mRBS (*p* < 0.001). The correlation between personality groups and ridden behavior scores helps further validate the scale composed by Ellis et al. [8] by putting it to the test with different grouping strategies. Although there is a possibility of scoring bias, all scoring was performed by one person. Additionally, the assigned personalities and ridden scores paralleled the observations provided by the riders. There were no statistical differences between modalities concerning mRBS (*p*: 0.271), suggesting that the modality is not related to the ridden behavior (Figure 2).

Lessons had the lowest overall mean mRBS (3.0) and the lowest mRBS scores. The lower mean and median mRBS of the lesson sessions could be influenced by the predominance of slower horses which are generally given to students. Horses with higher mRBS, around 4.5 and 5.0, are predominantly seen in the progression training sessions, including an outlier at 5.0; difficult horses were handled by the experienced riders. The interquartile range of the PT sessions is smaller than any other modality, showing that in 50% of the sessions, the mRBS was in the narrow ideal ridden behavior score range of 3.0 to 3.5, demonstrating a possible effect of greater rider experience in contrast to students who might avoid stimulating slow horses to move faster. The larger interquartile range of Lu, compared to PT, might be due to the lower effectiveness of indirect cues or a permissive attitude from the handler to allow the horse to express itself on the lunge line, or both. There was no significant correlation between mRBS and mIRTmax_a (ρ: 0.018; *p*: 0.883) and there were no statistical differences between personality groups regarding mIRTmax_b (*p*: 0.514). Personality and mRBS had no effect on infrared readings. Lastly, the study of mRBS and personality by age and sex yielded the following results: both sex and age correlated with mRBS (ρ: −0.487, *p* < 0.001 and ρ: −0.533, *p* < 0.001, respectively). This shows that an increase in age represents a general decrease in mRBS scores. Stallions (categorized as group 1 (mares: group 2; geldings: group 3) also registered higher mRBS. Geldings were statistically different from both mares and stallions regarding mRBS during exercise (*p* < 0.001), while the latter were identical. In general, younger horses, regardless of sex, had higher mRBS than older horses. In regard to personality, only sex showed a correlation (ρ: −0.488; *p* < 0.001), a similar correlation to mRBS and sex.

### 3.2. Workload Assessment

In total, 70 individual sessions were analyzed. Statistical differences in median total training time between modalities were found (*p* ˂ 0.001). The post hoc results show that lessons were statistically different from both lunging and progression training sessions (Figure 3).

Lessons were the longest sessions (52.4 ± 14.8 min), followed by lunging (35.0 ± 13.3 min) and progression training (27.5 ± 5.5 min). It is clear that PT had the smallest time range; correspondingly, the median was 27.0 min. No differences were found in IRTmax before and after a workout in any modality (Table 3). The results also show no statistical difference in mean ranks of IRTmax between the modalities before and after the workout (*p*: 0.073 and *p*: 0.082, respectively).

## 4. Discussion

### 4.1. Personality Assessment

The grouping strategy used in this study successfully predicted ridden behavior. Deviations from the predicted mRBS based on personality type could not be fully explained based on behavioral observations alone, whilst physiological markers could have uncovered reasons [8], such as pain or long-term stress. In general, the results suggest that horses display behaviors that are in accordance with their assigned personality type, regardless of the modality of training. It was expected that a horse would demonstrate higher mRBS behaviors during lunging and at higher frequencies, given that the horse is free of load and direct cues and, thus, seemingly freer to buck, kick and ignore cues. That was not the case in this study. The large range of mRBS in Lu demonstrates permissibility, but not a higher frequency of high mRBS. In fact, the only major deviation from the predicted mRBS was the outlier in PT sessions. The rider’s experience and riding style seem to influence ridden behavior, particularly in slower horses, by either demanding more activity or giving in to their behavior. Similar conclusions on the influence of rider behavior and riding style have been reported by other studies [37,38].

There was no relationship between eye surface temperature measured through IRT and personality or mRBS, which is in accordance with the results of previous studies [10,12]. In one study, the horses were divided into two groups based on whether the animal was compliant or not during the clipping procedure. During the sham clipping test, there was significantly increased activity in the non-compliant group versus the compliant group, and although both groups registered an increase in eye temperature, there was no difference between the groups until the very end of the procedure [10]. In a different study, the subjects were not grouped but submitted to two novel-stimulus tests where completion time and proactivity were recorded. In summary, high-strung personalities and their consequential behaviors were not mirrored in IRT imaging of the eye after tests [12]. In fact, no differences between proactivity percentages or groups were registered pre-test either [10,12], supporting the lack of relationship between personality groups and mRBS based on the IRT readings of the present study.

The lack of a relationship between personality type and age raises the question of whether this is due to habituation’s effect on the exercise process. Habituation is a form of learning in horses where the animal stops responding to frequent stimuli [2]. The workouts and all they entail are newer to young horses than stall and groundwork interaction with humans. A young horse that seems calm in a familiar stall when reacting to a new stimulus may showcase high mRBS scores during training because it can also fight the stressor to which it has not been habituated yet. In this study, this concept can be demonstrated by the particular case of one mare of personality type 2, aged 5 years, displaying the highest mRBS values, while other type 2 horses, aged over 10 years, scored much lower.

### 4.2. Workload Assessment

The largest deviation in total training time was registered in L sessions, which were expected because of the numerous student rider–horse pairs, while the same two riders performed PT within a working schedule. No differences in IRT readings were expected between modalities before workouts as there should not be a reason for it related to the modality, since all horses were handled identically until the moment of IRT imaging. The variation in eye temperature between the minimum and maximum values replicate previous studies. The standard deviation in this study (of 1.85 to 2.05 in absolute values) is relatively larger than those found in previous studies (about 0.09 to 0.9) [32,39], but it is not unheard for other studies to also reached large deviations over 1 [40]. Before-and-after eye temperature differences in this study did not reach 0.5 °C. Previous studies have found significant increases in eye surface temperature at values starting around 0.5 °C, [31,40] but more often above 0.7 °C [39,41,42] up to 1.63 °C [32].

In a previous study with young Friesian horses, the authors found that most horses performing a SET of lower effort than their regular training reached their anaerobic threshold when cantering continuously for four minutes [43]. The authors attributed this to two factors: the biomechanics of the breed’s gaits and, possibly, the muscle fiber composition [43]. It is also clear that over time, the workload of the same SET protocol lowers as the horses gain fitness [43]. In our study, not only were all horses older than 5 years old (and older than the Friesian horses of that study), but all were Lusitano or Luso crossbreeds. Older horses that had spent more time in training had developed their fitness levels and should be less prone to reaching their anaerobic threshold when performing the same SETs. On the other hand, Friesian horses have been noted to reach their anaerobic threshold faster than warmbloods [43,44], but no study with Lusitano horses in dressage training has tested for any specificity in this regard. This study missed the opportunity to introduce SETs as a standardizing tool. However, the aim was not to test increases in fitness level over time, but to test for differences between training methods. Mean HR recordings during dressage training have been between 62 bpm and 160 bpm [20,44,45,46], with HR during canter reaching nearly 180 bpm [43,44]. The generally accepted anaerobic threshold based on HR starts at around 200 bpm, leading to the conclusion that dressage is, for trained horses, a submaximal anaerobic effort [20]. Any measurable workload variable is expected to be significantly different from the results obtained in studies with race horses on treadmills or warmbloods and Thoroughbreds during cross-country or show-jumping training and showing, including the IRT readings. Trindade et al. [31] also registered an increase in eye temperature in ranch horses after a working day and considered the workload to be submaximal based on the increase in all fitness variables (maximum eye surface temperature, respiratory rate, plasma lactate concentrations, enzymatic activity of creatine kinase, serum concentration of total protein and plasma cortisol concentration) in correlation with workload variables (heart rate, duration, distance and average velocity); this resulted in a prolonged, slow-paced exercise, producing mostly aerobic energy [31]. The authors used the duration of the working day as the variable to categorize effort/workload (after putting aside distance), which presented much greater values (in hours) than the duration variable (in minutes, less than an hour) of the present study. It would be fair to conclude that the duration of the submaximal effort is key to understanding why the two studies present different outcomes. The discipline of endurance requires a combination of stamina and fitness for the horse to ride for kilometers (80 km to 160 km at international levels) for hours against the clock, with mandatory rests. Like dressage [20] and ranch work [31], endurance riding is an aerobic effort, as demonstrated by the HR and blood lactate concentrations of qualified horses [47].

The impossibility of performing testing indoors poses the question of whether the weather influenced our readings. In the present study, the mean air temperature was 16 °C (16.3 ± 3.8 °C), well within the 5 °C to 20 °C thermoneutral comfort zone of horses [48] and the 20 °C to 25 °C threshold considered by other papers [25,26]. On the other hand, the mean wind speed was 18,3 km/h (18.3 ± 5.3 km/h). On some days, we registered only light breezes (42% of days), and others had constant wind (29%) and even strong gusts (29%)—a major issue when performing IRT tests outdoors. Wind facilitates heat dissipation [25], and in addition, the cool-down time was 3 to 5 min on average, which might have further skewed the IRT readings by allowing the horse to cool down before the thermogram was taken. Previous studies registered that eye temperature decreased to pre-test values at the markers of 14 min [49] and 15 min [10] after a stressor was applied, but changes were registerable in the timespan of 5 min [10,50]. In a study with Thoroughbreds performing a high-speed treadmill exercise indoors, thermograms were taken at minutes 0, 5 and 15 of the exercise, and then, at minutes 45 and 60 post-exercise, and increased temperature was registered up to the marker of 15 min; however, it is uncertain at what point the body surface temperature stabilized [35]. In the present study, not only was the time the horses were exposed to wind considerable, but the cool down time might also have allowed for changes in eye temperature to take place before the IRT readings. In other studies, 5 min marks were established for practical reasons. Had the present study tested more biomarkers for fitness, the influence of climate conditions could have been clearer. Blood sampling and heart rate measurements are easily performed in the field [20]. The limitation of this study does not lie in whether it would be feasible, but rather, in the lack of HR reading equipment, of which a minimum of three would have been required given the number of individual sessions occurring simultaneously; moreover, this study was limited by the number of times a horse would be sampled over the course of the testing period, which could be up to 4 times a week, introducing a great level of stress [30] and discomfort.

Finally, the horses were tested in a familiar environment which they had all inhabited and trained on for at least a year. Previous studies were conducted to assess the effect of the competition environment as a stressor in horses. These included not the only warm up and the performance, but also the effect of the age of the animal, its breed, its stud lineage, the day of the competition, the time of sampling, the rider, the number of previous competitions, daily hours of training, journey duration, the type of transportation and the arrival date [45,46,47]. Therefore, despite studies such as Sánchez et al. [47] focusing on dressage-trained and -performing horses, the key difference between that study and the present one is the competitive, non-familiar environment.

These results on their own cannot uphold the hypothesis that lunging is a form of lower-impact training compared to a ridden workout as no differences were found at all, or the hypothesis that longer sessions impact the horse equally to high-frequency, shorter sessions. In fact, with no statistical differences in any modality between pre- and post-workout readings, drawing conclusions between the modalities of training is not feasible. New studies that aim to test IRT in training horses ought to avoid the same constraints as those of this study, mainly increasing the number of subjects, the possibility of performing testing indoors and the introduction of more biomarkers as comparable variables.

## 5. Conclusions

Behavior during a workout, regardless of modality, was shown to be a reflection of the tested personality, sex and age of horses. However, neither personality nor ridden behavior were related to workload, as no significant changes were determined between the pre- and post-workout readings. Although IRT has been tested with clear results in dressage horses in competition, the results of this study could not demonstrate the usefulness of this tool in assessing workload in dressage horses in training and in a familiar environment. On the other hand, the results could demonstrate that overworking did not occur, but cannot set aside the possibility of underworking. Both conclusions also apply to lunging work. New studies are warranted to test IRT in dressage horses training in a familiar environment, which ought to incorporate other easy-access markers such as HR and velocity; these are available on the market specifically for equestrians. Finally, more conclusive studies on workload during lunging are also necessary for a better understanding of the process and impact of lunging sessions.

## Figures and Tables

**Figure 1 animals-12-03255-f001:**
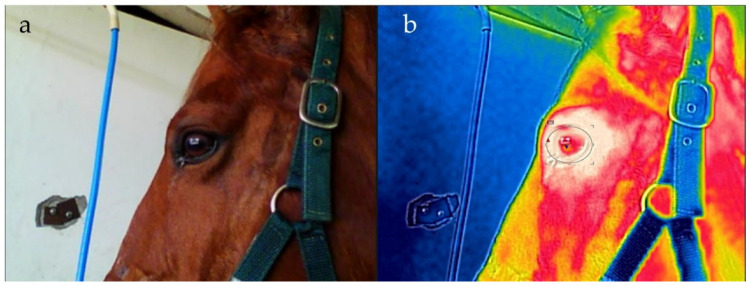
Image of a horse head (**a**) and infrared thermogram of the head of the same horse showing the ellipse fitted to the eye to determine eye temperature (**b**).

**Figure 2 animals-12-03255-f002:**
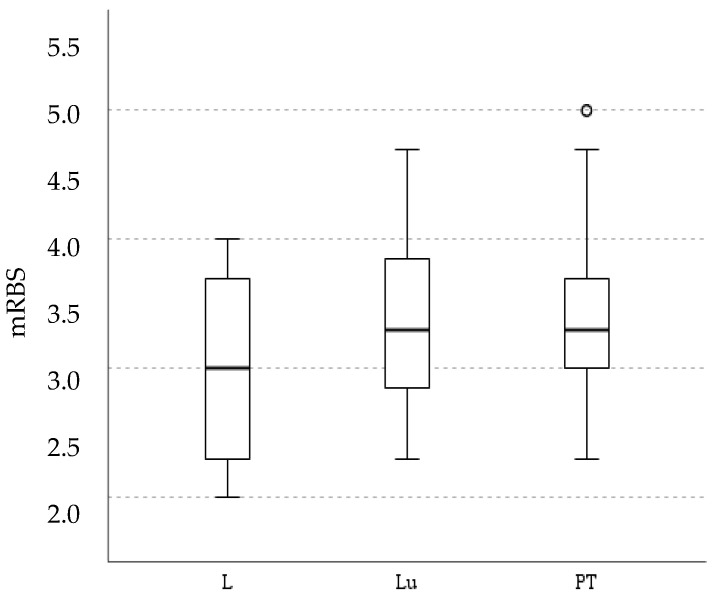
Boxplot chart for mRBS (mean of the three ridden behavior scores assigned in a session) values across the modalities of training—lessons (L), lunging (Lu) and progression training (PT).

**Figure 3 animals-12-03255-f003:**
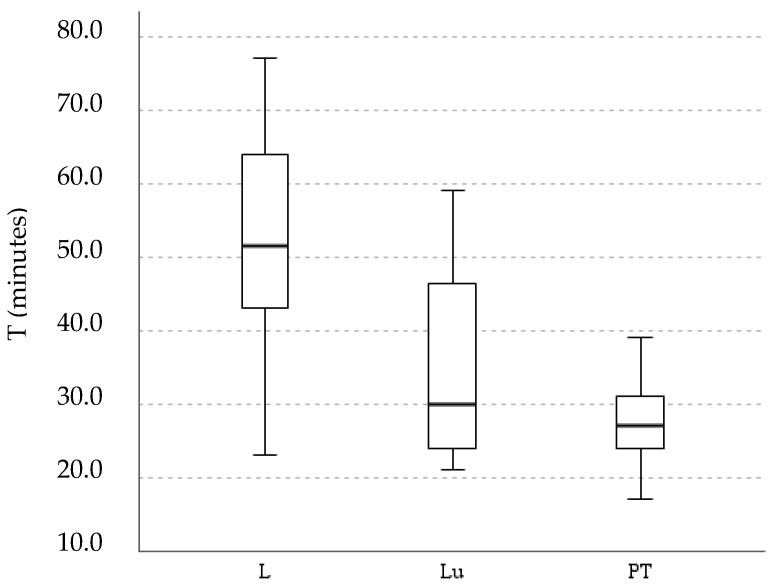
Boxplot chart for T (in minutes) across the modalities of training—lessons (L), lunging (Lu) and progression training (PT).

**Table 1 animals-12-03255-t001:** Reactivity behavior scale used during novel-stimulus test.

Reactivity Score	Descriptors
1	Inspects the object, no reaction to position, ears and eyes turn to the object, returns to normal activity.
2	Inspects the object, reacts to position, ears and eyes turn to the object, limits normal activity.
3	Avoids the object, head high and away from object, fixated on the object, walks in circles, interrupts normal activity.

**Table 2 animals-12-03255-t002:** Ridden horse behavior scale.

Reactivity Score	Descriptors
1	Slow, requires constant aid to move forward, falls back on trot/canter, falls back on exercises.
2	Relaxed, requires aids to move forward, falls back on exercises.
3	Relaxed, good response to aids, moving forward and performing exercises.
4	Slightly aroused, rushing paces and exercises, jumpy responses to aids.
5	Agitated, threatens to bolt, light bucks and erratic movements, refuses exercises.
6	Highly agitated, bolting, rearing or bucking.

**Table 3 animals-12-03255-t003:** Descriptive analysis of IRTmax, in degrees Celsius, within modalities of training—lessons (L), lunging (Lu) and progression training (PT).

	Mean ± SD		Significance Value		Minimum		Maximum	
	Before	After	Z	*p*-Value	Before	After	Before	After
L	35.42 ± 1.86	35.62 ± 1.90	Z: −0.004	*p*: 0.689	31.63	31.64	38.85	38.84
Lu	34.67 ± 2.05	34.83 ± 1.97	Z: −0.661	*p*: 0.508	31.07	31.20	38.40	38.35
PT	35.40 ± 1.90	35.43 ± 1.85	Z: −0.035	*p*: 0.972	31.60	31.65	39.12	39.14

## Data Availability

The data presented in this study are available on request from the corresponding author.

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
