# Peer review of "Use of Infrared Thermography to Assess Body Temperature as a Physiological Stress Indicator in Horses during Ridden and Lunging Sessions"

_animals, 2022, doi:10.3390/ani12233255_

Round 1

Reviewer 1 Report

Just a few comments:

1) It would be helpful  to have a statistician whose native language is English to review for correct language regarding results and conclusions. For example, the words "effect" and "affected by" are inappropriate for the methods used which result in associations rather than effects (in abstracts, simple summaries, and the paper). Also, care should be taken when saying "no association" with such small numbers of animals and so many categories with very few animals in each, unless you have done a power analysis and can show that there was sufficient power to detect a difference.

2) In the title, suggest using "physiological stress indicator",  to distinguish from psychological stress.

3) Since you report no difference in eye temperature before and after exercise, and conclude no physiological stress in any of these horses in any exercise, it would be helpful for you to discuss how sensitive is thermography as an indicator of physiological stress or fitness? What type of rise in temperature would correlate with other measures of physiological stress.

4) I think it is a bit of a stretch to categorize personality types simply on novel object test.

Author Response

Reviewer #1

Comments and Suggestions for Authors

Just a few comments:

1) It would be helpful  to have a statistician whose native language is English to review for correct language regarding results and conclusions. For example, the words "effect" and "affected by" are inappropriate for the methods used which result in associations rather than effects (in abstracts, simple summaries, and the paper). Also, care should be taken when saying "no association" with such small numbers of animals and so many categories with very few animals in each, unless you have done a power analysis and can show that there was sufficient power to detect a difference.

Response: Thank you for pointing this out. We have ask for help of other colleague and the text was reviewed.

2) In the title, suggest using "physiological stress indicator",  to distinguish from psychological stress.

Response: The title was changed. Now ““Use of Infrared Thermography to assess body temperature as a physiological stress indicator on horses during ridden and lunging sessions”

3) Since you report no difference in eye temperature before and after exercise, and conclude no physiological stress in any of these horses in any exercise, it would be helpful for you to discuss how sensitive is thermography as an indicator of physiological stress or fitness? What type of rise in temperature would correlate with other measures of physiological stress.

Response: We have taken your input into account and elaborated further on this specific topic.

4) I think it is a bit of a stretch to categorize personality types simply on novel object test.

Response: We appreciate the reviewer’s input.

Reviewer 2 Report

The search for use of Infrared Thermography to assess body temperature as a stress indicator on horses during ridden and lunging sessions is of great interest in the practice of sports medicine of this species. Current training methods might be efficient in the short term but lack scientific evidence that disproves any undesired effects it might have on the horse, both physically and emotionally. The present study was to evaluate the usefulness of Infrared thermography before and after sessions to determine maximum eye surface IRT temperature (IRTmax), and total training time. In addition, the novel stimuli test was conducted for personality assessment, and the ridden behavior was scored. Authors concluded that trained horses show little stress when working in a familiar environment and the workout plan was submaximal. The personality test was adequate and positively correlated to the ridden behavior. However, the main limitation of the study is low number of horses and different age of them. Thus, the experience level of the horses varies. More recent citations should be added mostly those connected with morse fitness monitoring.

Specific comments

Introduction

L57: The authors are calculating that several parameters can be measured for evaluate the horses fitness such as heart rate (HR), ventilation rate (VR), and blood and plasma lactate concentration. However, they are not mentioning that Infrared Thermography was used for that as well because it correlates with lactate concentration in race horses. Thus, this findings should be added. In addition several other factors has been measured to obtain the horses fitness such as changes in serum amyloid A, creation of anti-inflammatory state or the physical activity-dependent hematological and biochemical changes.

Materials & methods

L86: is it median or mean for the age? It should be clarified. The stable conditions should be added as well as weather conditions when the examination was performed because it influence on IRT evaluation.

L116: the duration of walking, trotting and canter on both leads should be added.

L120: the riders were different in all horses?

L122: The lunging person was different for all horse?

Results

Figure 2 – please add what is it mRBS

Table 3 – add the p value

Discussion

L222 – It is very interesting that horses would not demonstrate higher mRBS behaviors during lunging. However, it may be connected to with high level exploitation during riding session. How long was the break between training and lunging? However, there was several publications about stress response during the training and optimal training is not leading to stress which may be influential on kicking and ignore cues behavior to destress.

L226 – was the ridder’s experience measured in this study? If yes, it will be interesting to add such data.

L249 – such information such be added to materials and methods section as well.

L324 – the study limitation should be added. The low number of horses is one of them however I am aware that not always is possible to collect more animals during in vivo study. However, the experience level of the horses varies. In addition, some markers such as HR recordings or cortisol evaluation (stress hormone) will be helpful but once again not always is possible to perform invasive examination. However, such information should be added.  

Conclusion

This part should be more comprehensive. Thus, the Authors should in brief conclude their findings.

Thus, I recommend this article for publication after minor corrections.

Author Response

Reviewer #2

Comments and Suggestions for Authors

The search for use of Infrared Thermography to assess body temperature as a stress indicator on horses during ridden and lunging sessions is of great interest in the practice of sports medicine of this species. Current training methods might be efficient in the short term but lack scientific evidence that disproves any undesired effects it might have on the horse, both physically and emotionally. The present study was to evaluate the usefulness of Infrared thermography before and after sessions to determine maximum eye surface IRT temperature (IRTmax), and total training time. In addition, the novel stimuli test was conducted for personality assessment, and the ridden behavior was scored. Authors concluded that trained horses show little stress when working in a familiar environment and the workout plan was submaximal. The personality test was adequate and positively correlated to the ridden behavior. However, the main limitation of the study is low number of horses and different age of them. Thus, the experience level of the horses varies. More recent citations should be added mostly those connected with more fitness monitoring.

Response: We would like to thank the reviewer for his/her positive and insightful comments on the manuscript.

Specific comments

Introduction

L57: The authors are calculating that several parameters can be measured for evaluate the horses fitness such as heart rate (HR), ventilation rate (VR), and blood and plasma lactate concentration. However, they are not mentioning that Infrared Thermography was used for that as well because it correlates with lactate concentration in race horses. Thus, this findings should be added. In addition several other factors has been measured to obtain the horses fitness such as changes in serum amyloid A, creation of anti-inflammatory state or the physical activity-dependent hematological and biochemical changes.

Response: We have now included previous studies that have successfully correlated IRT to other biomarkers of fitness evaluation.

Materials & methods

L86: is it median or mean for the age? It should be clarified. The stable conditions should be added as well as weather conditions when the examination was performed because it influence on IRT evaluation.

Response: We thank the reviewer for noticing such omission. We have clarified what was asked. Stabling conditions and a summary of weather conditions have been added.

L116: the duration of walking, trotting and canter on both leads should be added.

Response: We understand now that a key piece of information was left out – the total of individual sessions analyzed was 70. We added this information to the text as well because we believe it helps clarify decisions taken during the study. Although we do have the durations of walk, trot and canter for every single session, and it is possible to compile all the information into a table of means, maximums and minimums, we had decided to only consider total training time.

L120: the riders were different in all horses?

L122: The lunging person was different for all horse?

Response: We are not very clear in our writing concerning riders. We do recognize this and so added more information in text to help clarify. In total there were 7 riders (5 students eligible for this study and the 2 staff riders). The riders were not different in all horses: staff riders rode all horses in PT sessions and student riders rotated rides frequently as per the rules of the trainer.

Results

Figure 2 – please add what is it mRBS

Response: The information required was added.

Table 3 – add the p value

Response: Although the p-value was in table 3, we separated z and p-value into different columns to make it more evident and of easier read.

Discussion

L222 – It is very interesting that horses would not demonstrate higher mRBS behaviors during lunging. However, it may be connected to with high level exploitation during riding session. How long was the break between training and lunging? However, there was several publications about stress response during the training and optimal training is not leading to stress which may be influential on kicking and ignore cues behavior to destress.

Response: Our IRT and behavior results don’t suggest a high level of exploitation. The intervals between ridden training and lunging was every other day (or lunging after two days of ridden training). We didn’t include horses in the breaking phase, but we do recognize that lunging training requires further studying and in our study we could have compared the behavior of trained horses to those in breaking when lunged to further develop the discussion on this matter.

L226 – was the ridder’s experience measured in this study? If yes, it will be interesting to add such data.

Response: Rider experienced was not measured in this study. We are aware of its important and considered rider experience in our discussion.

L249 – such information such be added to materials and methods section as well.

Response: If we are correct in assuming that line 249 refers to the sentence below, we are not quite sure what the reviewer is asking for. In subsection 2.5.1. of the materials and methods section, we preface the statistical tests and grouping strategies that lead to the results and conclusions of line 249.

“In this study this concept can be demonstrated by the particular case of one personality type 2 mare, age 5 years, displaying the highest mRBS values while other type 2 horses, ages over 10 years, scored much lower.”

L324 – the study limitation should be added. The low number of horses is one of them however I am aware that not always is possible to collect more animals during in vivo study. However, the experience level of the horses varies. In addition, some markers such as HR recordings or cortisol evaluation (stress hormone) will be helpful but once again not always is possible to perform invasive examination. However, such information should be added.  

 Response: We recognize the many limitations of our study and heeded to your suggestion to address this topic. We discuss our study limitations further in the text and included a final note for future studies.

Conclusion

This part should be more comprehensive. Thus, the Authors should in brief conclude their findings.

Response: The conclusion section was revised